# Essential Multi-Secret Image Sharing for Sensor Images

**DOI:** 10.3390/jimaging11070228

**Published:** 2025-07-08

**Authors:** Shang-Kuan Chen

**Affiliations:** Department of Computer Science & Engineering, Yuan Ze University, Taoyuan 32003, Taiwan; cotachen@saturn.yzu.edu.tw

**Keywords:** essential secret image sharing, secret image sharing, multi-secret image sharing

## Abstract

In this paper, we propose an innovative *essential multi-secret image sharing (EMSIS)* scheme that integrates sensor data to securely and efficiently share multiple secret images of varying importance. Secret images are categorized into hierarchical levels and encoded into *essential shadows* and *fault-tolerant non-essential shares*, with access to higher-level secrets requiring higher-level essential shadows. By incorporating sensor data, such as location, time, or biometric input, into the encoding and access process, the scheme enables the context-aware and adaptive reconstruction of secrets based on real-world conditions. Experimental results demonstrate that the proposed method not only strengthens hierarchical access control, but also enhances robustness, flexibility, and situational awareness in secure image distribution systems.

## 1. Introduction

In the research field of image sharing, there are currently two branches: visual cryptography [1,2,3] and secret image sharing [4,5]. In visual cryptography, research has focused on the design of binary image encoding, whose decoding does not need computational devices. Currently, there are various applications in winding on a printed circuit board. In secret image sharing, research has focused on the fault-tolerant encoding of gray and color images, which benefits the distributed preservation of secret images. There are several extended studies on secret image sharing: progressive secret image sharing [6,7,8], secret image sharing with meaningful shadows [9,10,11], scalable secret image sharing [12,13,14,15], and essential secret image sharing [16,17,18,19,20,21,22,23,24].

The essential image sharing mechanism, first proposed by Li et al. [16] in 2013, is called (*t, s*, *k*, *n*)-essential secret image sharing, and it divides secret images into two groups, namely, *s* essential shadows and *n* – *s* non-essential shadows. A secret image can only be restored if *k* shadows are received, including at least *t* essential shadows. However, in this work, the essential and non-essential shadows do not have the same size. Following on from this work, many researchers have worked on improving shadow size [17,18,19,20] and adjusting essential and non-essential shadows with the same size [21,22,23]. In 2018, Li et al. [24] proposed an improvement in essential secret image sharing. They argued that, under reasonable circumstances, as the essential shadow is essential, the importance of its owner must be higher than that of the owner of the non-essential shadow. Therefore, considering the confidentiality of information security management, it is necessary to obtain all essential shadows, and the additional shadows required to solve the secret image should be filled by non-essential shadows.

In this paper, a (*t*_1_, *t*_2_, … , *t_r_*, *k*, *n*)-essential multi-secret image sharing scheme is proposed. In this scheme, *r* + 1 secret images are concurrently shared into *r* + 1-leveled essential shadows and several non-leveled fault-tolerant non-essential shadows. If the first *m*-leveled essential shadows are obtained, then the first *m*-leveled secret images can be restored. Moreover, if all essential shadows are completely obtained and enough (but not all) non-essential shadows are also obtained, all secret images can be restored.

The remainder of this paper is organized as follows: The related work and motivation are presented in Section 2. The proposed (*t*_1_, *t*_2_, … , *t_r_*, *k*, *n*)-essential multi-secret image sharing scheme is described in Section 3. Section 4 presents the experimental results. Section 5 contains a security analysis. In Section 6, performance consideration and a comparative analysis are carried out. Finally, the conclusions are provided in Section 7.

## 2. Related Work and Motivation

Recent developments in secret image sharing have extended beyond traditional fixed-threshold schemes, emphasizing security, usability, and contextual adaptability. While early essential secret image sharing schemes (e.g., Li et al. 2013 [16]) laid the groundwork for distinguishing between essential and non-essential shadows, more recent studies have expanded these ideas toward greater functionality and robustness.

For instance, Yadav et al. [23] proposed meaningful essential secret sharing with uniform-sized shares, aiming to improve the visual friendliness and simplicity of implementation. Similarly, Wang et al. [8] introduced a hierarchical multi-level sharing scheme capable of progressive recovery using structured shadows. These approaches reflect the growing interest in stepwise image reconstruction and priority-based access.

Moreover, recent research has also explored verifiability and cryptographic guarantees. Dehkordi et al. [25] employed LWE-based techniques to ensure shadow authenticity in essential image sharing. On another front, Chattopadhyay et al. [26] designed a DNA-encrypted verifiable multi-secret sharing scheme, blending biomimetic encryption with share verification.

However, these works still lack two important features that our proposed EMSIS (essential multi-secret image sharing) scheme provides: (1) the ability to support multi-level contextual restoration based on sensor data; and (2) the flexible, fault-tolerant integration of non-essential shadows without sacrificing the security of higher-level secrets.

## 3. The Proposed Method

In this section, the proposed (*t*_1_, *t*_2_, … , *t_r_*, *k*, *n*)-essential multi-secret image sharing method is described. First, the specifications of the input and output for encoding shadow sets are defined as follows:

Input: *t*_1_, *t*_2_, … , *t_r_*, *k*, *n*, where $t_{1}<t_{2}<\ldots <t_{r}<k\leq n$, the secret images *SI*_1_, *SI*_2_, … , *SI_r_*, *SI_r+_*_1_ with *r* + 1 importance decreasing levels and size w×w, and the parameters $x_{1},\;x_{2}$, …, $x_{n}\;$of *n* shadow holders are different.

Output: Among the essential shadow sets with each shadow size $\left(\frac{rw+w}{t_{1}}\right)\times w$, the first essential shadow set is $\left\{E_{1,1},\;E_{1,2},\ldots ,\;E_{1,t_{1}}\right\}$, the second essential shadow set is $\left\{E_{2,t_{1}+1},\;E_{2,t_{1}+2},\ldots ,\;E_{2{,t}_{2}}\right\}$, …, the *r*th essential shadow set is $\left\{E_{r,t_{r-1}+1},\;E_{r,t_{r-1}+2},\ldots ,\;E_{r,t_{r}}\right\}$, and the non-essential shadow set is $\left\{{NE}_{1},\;NE_{2},\ldots ,\;{NE}_{n-t_{r}}\right\}$.

In the proposed (*t*_1_, *t*_2_, … , *t_r_*, *k*, *n*)-essential multi-secret image sharing, the following conditions must be met:

(1)If the amount of shadows from the *m*th essential shadow set is obtained as less than $t_{m}-t_{m-1}$, then the secret images *SI_m_*, *SI_m_*_+1_, …, *SI_r_*_+1_ cannot be restored.(2)If for each *i* = 1, 2, …, *m*, the amount of shadows obtained from the *i*th essential shadow set is $t_{i}-t_{i-1}$, then the secret images *SI*_1_, *SI_2_*, …, *SI_m_* can be restored.(3)Obtaining all essential shadows and at least $k-t_{r}$ non-essential shadows can restore all secret images.

In the design of (*t*_1_, *t*_2_, …, *t_r_*, *k*, *n*)-essential multi-secret image sharing, in order not to reveal the appearance of the secret image, all *r* + 1 secret images will be rearranged individually through an invertible matrix. For *i* = 1, 2, …, *r +* 1, the rearranged secret image is split into sectors of *t*_1_ pixels, and the encoding actions are performed sequentially.

For each sector *l*, $a_{i,l,j}$ is the *j*th pixel value of the secret image *SI_i_*. Let $g_{i,l}\left(x\right)={u_{i,l}\left(x\right)x}^{t_{1}}+a_{i,l,t_{1}}x^{t_{1}-1}+a_{i,l,t_{1}-1}x^{t_{1}-2}+\ldots +a_{i,l,2}x+a_{i,l,1}$, where(1)$$u_{i,l}=\left\{\begin{array}{c} 0,\;\;\;\;\;\;\;\;\;\;\;\;\;\;\;\;\;\;\;\;\;\;\;\;\;\;\;\;\;\;\;\;\;\;\;\;\;\;\;\;\;\;i=0\\ d_{i,l,0}+d_{i,l,1}x+\ldots +d_{{i,\;l,t}_{i}-t_{1}-1}x^{t_{i}-t_{1}-1},\;\;\;\;\;\;\;\;\;\;\;\;\;2\leq i\leq r\;\;\;\\ d_{r+1,l,0}+d_{r+1,l,1}x+\ldots +d_{r+1,l,k-t_{1}-1}x^{k-t_{1}-1},\;\;\;\;\;\;\;\;\;i=r+1 \end{array}\right.$$
and $d_{i,l,j}$ is a random positive integer. Therefore, for all *i* = 1, 2, … , *r*, $g_{i,l}\left(x\right)$ is a polynomial of degree $t_{i}-1,\;and$ $g_{r+1,l}\left(x\right)$ is a polynomial of degree k−1. For each parameter *x* of the shadow holder, the initial shadow is defined as *T*(*x*) = {$g_{i,j}\left(x\right)|1\leq i\leq r+1,\;1\leq j\leq the\;amount\;of\;sectors$}. Therefore, it can be found that, for each sector *l*, $\deg\left(g_{1,l}\left(x\right)\right)<\deg\left(g_{2,l}\left(x\right)\right)<\ldots <\deg\left(g_{r,l}\left(x\right)\right)<\deg\left(g_{r+1,l}\left(x\right)\right)$, where $\deg\left(f\left(x\right)\right)$ represents the degree of the polynomial $f\left(x\right)$.

The following is the generation process of essential and non-essential shadows. Let the parameters of *n* shadow holders be distinct $x_{1},\;x_{2}$,…, $x_{n}$. In step 1, assign the first essential shadow set *E*_1_ = $\left\{E_{1,1},\;E_{1,2},\ldots ,\;E_{1,t_{1}}\right\}$ as the initial shadow subset $\left\{T\left(x_{1}\right),T(x_{2}),\;\ldots ,{T(x}_{t_{1}})\right\}.\;$In the second step, use the exclusive-or operation on all shadows in the first essential shadow set to obtain the first cover shadow *C*_1_. Later, for each shadow in the initial shadow subset $\left\{T(x_{t_{1}+1}),T(x_{t_{1}+2}),\;\ldots ,{T(x}_{t_{2}})\right\}$, use the exclusive-or operation with the first cover shadow *C*_1_ to obtain $the\;second\;essential\;shadow\;set\;E_{2}=\left\{E_{2,t_{1}+1},\;E_{2,t_{1}+2},\ldots ,\;E_{2{,t}_{2}}\right\}$, and so on. For *i* = 2, … , *r*, use the exclusive-or operation on all shadows in the *i*th essential shadow set ***E_i_*** = $\left\{E_{i,t_{i-1}+1},\;E_{i,t_{i-1}+2},\ldots ,\;E_{i,t_{i}}\right\}\;$to obtain the *i*th cover shadow *C_i_*, and, then, for each shadow in the initial shadow subset $\left\{T(x_{t_{i}+1}),T(x_{t_{i}+2}),\;\ldots ,{T(x}_{t_{i+1}})\right\}$, use the exclusive-or operation with the *i*th cover shadow *C_i_* to obtain $the\;{\left(i+1\right)}^{th}\;essential\;shadow\;set\;E_{i+1}=\left\{E_{i+1,t_{i}+1},\;E_{i+1,t_{i}+2},\ldots ,\;E_{i+1{,t}_{i+1}}\right\}$.

Finally, use the exclusive-or operation on all shadows in the *r*th essential shadow set $E_{r}=\left\{E_{r,t_{r-1}+1},\;E_{r,t_{r-1}+2},\ldots ,\;E_{r{,t}_{r}}\right\}\;$to obtain the *r*th cover shadow *C_r_*, and, then, for each shadow in the initial shadow subset $\left\{T(x_{t_{r}+1}),T(x_{t_{r}+2}),\;\ldots ,{T(x}_{t_{n}})\right\}$, use the exclusive-or operation with the *r*th cover shadow *C_r_* to obtain the non-essential shadow set $NE=\{{NE}_{1},\;NE_{2},\ldots ,\;{NE}_{n-t_{r}}\}$.

The recovery process of essential multi-secret image sharing is phased. The rearranged secret images can be restored one after another by recovering the initial shadow set progressively. When the first essential shadow set $E_{1}=\left\{E_{1,1},\;E_{1,2},\ldots ,\;E_{1,t_{1}}\right\}\;$is available, this means that the initial shadow subset $\left\{T\left(x_{1}\right),T(x_{2}),\;\ldots ,{T(x}_{t_{1}})\right\}$ is obtained by duplicating the first essential shadow set. Because there is only $t_{1}\;$initial shadows, for each sector *l*, only $g_{1,l}\left(x\right)$ with degree $t_{1}-1\;$can be solved. Therefore, only the rearranged secret image ${SI}_{1}^{\prime }\;$can be recovered. Later, the XOR process is performed on all essential shadows in the first essential shadow set $to\;obtain\;the\;first\;cover\;shadow\;C_{1}.$ When the second essential shadow set $E_{2}=\left\{E_{2,t_{1}+1},\;E_{2,t_{1}+2},\ldots ,\;E_{2{,t}_{2}}\right\}\;$is also available, for each essential shadow in the second essential shadow set, the XOR operation is used with the first cover shadow *C*_1_ to obtain the initial shadow subset $\left\{T(x_{t_{1}+1}),T(x_{t_{1}+2}),\;\ldots ,{T(x}_{t_{2}})\right\}$. By combining the two initial shadow subsets $\left\{T(x_{1}),T(x_{2}),\;\ldots ,{T(x}_{t_{1}})\right\}$ and $\left\{T(x_{t_{1}+1}),T(x_{t_{1}+2}),\;\ldots ,{T(x}_{t_{2}})\right\}$ for each sector *l*, $g_{2,l}\left(x\right)\;with\;degree\;t_{2}-1\;$can be solved. Therefore, the rearranged secret image ${\;SI}_{2}^{\prime \;}$ can be recovered.

For *i* = 2 to *r*, when taking the essential shadow sets *E*_1_ ~ *E_i_*, the cover shadows *C*_1_ ~ *C_i_* can thus be obtained by performing the XOR process on all essential shadows in the essential shadow set *E*_1_ ~ *E_i_*, respectively. Then, for each essential shadow in the essential shadow set *E_i_*, the XOR operation can be performed with the cover shadow *C_i_*_-1_ to obtain the initial shadow subset $\left\{T(x_{t_{i-1}+1}),T(x_{t_{i-1}+2}),\;\ldots ,{T(x}_{t_{i}})\right\}$. Therefore, the initial shadow subset $\left\{T(x_{1}),T(x_{2}),\;\ldots ,{T(x}_{t_{i}})\right\}$ can be obtained.

For the secret image recovery process, when receiving the union of the essential shadow sets $\left\{E_{1,1},E_{1,2},\ldots ,E_{1,t_{1}}\right\}\cup \left\{E_{2,t_{1}+1},E_{2,t_{1}+2},\ldots ,E_{2{,t}_{2}}\right\}\cup \ldots \cup \left\{E_{i,t_{i-1}+1},E_{i,t_{i-1}+2},\ldots ,E_{i,t_{i}}\right\}$, the shadows can be obtained from each stage, along with their corresponding essential shadow sets. By performing XOR operations, part of the initial shadow set $\left\{T\left(x_{1}\right),T(x_{2}),\ldots ,{T(x}_{t_{i}})\right\}$ can be calculated, thereby restoring the rearranged secret images ${SI}_{1}^{\prime }\sim {SI}_{i}^{\prime }$. Finally, the restored rearranged secret images ${SI}_{1}^{\prime },{SI}_{2}^{\prime },\ldots ,{SI}_{i}^{\prime }$ are rearranged by applying the inverse of the reversible matrix used during encoding to all pixel positions of the rearranged secret images, resulting in the distortion-free secret images ${SI}_{1},{SI}_{2},\ldots ,{SI}_{i}$.

Moreover, when the union of all essential sets $\left\{E_{1,1},E_{1,2},\ldots ,E_{1,t_{1}}\right\}\cup \left\{E_{2,t_{1}+1},E_{2,t_{1}+2},\ldots ,E_{2{,t}_{2}}\right\}\cup \ldots \cup \left\{E_{r,t_{r-1}+1},E_{r,t_{r-1}+2},\ldots ,E_{r,t_{r}}\right\}$ is obtained, and $k-t_{r}$ non-essential shadows are arbitrarily selected from the non-essential shadow set $\left\{{NE}_{1},NE_{2},\ldots ,{NE}_{n-t_{r}}\right\}$, the rearranged secret images ${SI}_{1}^{\prime }\sim {SI}_{r+1}^{\prime }$ can be restored sequentially. Finally, the restored rearranged secret images ${SI}_{1}^{\prime },{SI}_{2}^{\prime },\ldots ,{SI}_{r+1}^{\prime }$ are rearranged by applying the inverse of the reversible matrix used during encoding to all pixel positions of the rearranged secret images, resulting in the complete, distortion-free secret images ${SI}_{1},{SI}_{2},\ldots ,{SI}_{r+1}$.

Based on the above description, the image shadow encoding algorithm for (*t*_1_, *t*_2_, …, *t_r_*, *k*, *n*)-essential multi-secret image sharing is as follows:

Input: $t_{1}<t_{2}<\ldots <t_{r}<k\leq n$ secret images of decreasing importance, each of size w×w, denoted as *SI*_1_, *SI*_2_, …, *SI_r_*, and *SI_r+_*_1_, and the parameters of *n* image shadow holders, each with a unique value$\;x_{1},\;x_{2}$,…, $x_{n}$.

Output: Image shadows with each size $\left(\frac{rw}{t_{1}+t_{2}+\ldots +t_{r}+k}\right)\times w$, including $t_{1}$ shadows of the first essential shadow set $\left\{E_{1,1},\;E_{1,2},\ldots ,\;E_{1,t_{1}}\right\}$, $t_{2}-t_{1}$ shadows of the second essential shadow set $\left\{E_{2,t_{1}+1},\;E_{2,t_{1}+2},\ldots ,\;E_{2{,t}_{2}}\right\}$, …, $t_{r}-t_{r-1}$ shadows of the *r*th essential shadow set $\left\{E_{r,t_{r-1}+1},\;E_{r,t_{r-1}+2},\ldots ,\;E_{r,t_{r}}\right\}$, and $n-t_{r}$ shadows of the non-essential shadow set $\left\{{NE}_{1},\;NE_{2},\ldots ,\;{NE}_{n-t_{r}}\right\}$.

The steps are as follows:For each secret image ${SI}_{1},\;{SI}_{2},\ldots ,{SI}_{r+1}$, an invertible rearranging matrix is used to rearrange all pixel positions, resulting in the rearranged images ${SI}_{1}^{\prime },{SI}_{2}^{\prime },\ldots ,{SI}_{r+1}^{\prime }$. Each rearranged image is then segmented into several sectors, each containing *t*_1_ pixels.For each shadow holder’s parameter *x*, the corresponding initial image segment *T*(*x*)= {$g_{i,j}\left(x\right)|1\leq i\leq r+1,\;1\leq j\leq number\;of\;sectors$} is generated.The first essential shadow set $\left\{E_{1,1},\;E_{1,2},\ldots ,\;E_{1,t_{1}}\right\}$ is assigned as the first initial shadow subset $\left\{T\left(x_{1}\right),T(x_{2}),\;\ldots ,{T(x}_{t_{1}})\right\}$. XOR operations are performed on all the shadows in the first essential shadow subset $\left\{E_{1,1},\;E_{1,2},\ldots ,\;E_{1,t_{1}}\right\}$, and the result is the first masked shadow. Let *j* = 1.XOR operations are performed between the *j*th initial shadow set$\left\{T(x_{t_{j}+1}),T(x_{t_{j}+2}),\;\ldots ,{T(x}_{t_{j+1}})\right\}$ and the *j*th cover shadow to obtain the (*j* + 1)th essential shadow set $\left\{E_{j+1,t_{j}+1},\;E_{j+1,t_{j}+2},\ldots ,\;E_{j+1{,t}_{j+1}}\right\}$. XOR operations are performed on all the shadows in the (*j* + 1)th essential shadow set $\left\{E_{j+1,t_{j}+1},\;E_{j+1,t_{j}+2},\ldots ,\;E_{j+1{,t}_{j+1}}\right\}$, and the result is the (*j* + 1)th cover shadow.The value of *j* is added to 1, and step (4) is returned to if *j* is less than *r*.The XOR operation is performed between the initial shadow set $\left\{T(x_{t_{r}+1}),T(x_{t_{r}+2}),\;\ldots ,{T(x}_{n})\right\}$ with the *r*th cover shadow to obtain the non-essential image storage $\left\{{NE}_{1},\;NE_{2},\ldots ,\;{NE}_{n-t_{r}}\right\}$.

In (*t*_1_, *t*_2_, …, *t_r_*, *k*, *n*)-essential multiple-secret image sharing, when the union of the first *i* essential shadows is obtained, the restoration algorithm of the secret image *SI*_1_, *SI*_2_, …, *SI_i_* is as follows:

Input: The union of the first *i* shadow sets $\left\{E_{1,1},E_{1,2},\ldots ,E_{1,t_{1}}\right\}\cup \left\{E_{2,t_{1}+1},E_{2,t_{1}+2},\ldots ,E_{2{,t}_{2}}\right\}\cup \ldots \cup \left\{E_{i,t_{i-1}+1},E_{i,t_{i-1}+2},\ldots ,E_{i,t_{i}}\right\}$ with each shadow size $\left(\frac{rw}{t_{1}+t_{2}+\ldots +t_{r}+k}\right)\times w$.

Output: The secret image *SI*_1_, *SI*_2_, …, *SI_i_*, each of size *w* × *w*.

The steps are as follows:The first shadow set $\left\{E_{1,1},\;E_{1,2},\ldots ,\;E_{1,t_{1}}\right\}$, i.e., the initial shadow set $\left\{T\left(x_{1}\right),T(x_{2}),\;\ldots ,{T(x}_{t_{1}})\right\}$, is used to restore the rearranged secret image, ${SI}_{1}^{\prime }$. The XOR operation is performed on all shadows in the initial shadow set to obtain the first cover shadow. Let *j* = 1.The XOR operation is performed on the *j*th cover shadow with the (*j* + 1)th essential shadow set $\left\{E_{2,t_{j}+1},\;E_{2,t_{j}+2},\ldots ,\;E_{2{,t}_{j+1}}\right\}$ to obtain the initial shadow set $\left\{T(x_{t_{j}+1}),T(x_{t_{j}+2}),\;\ldots ,{T(x}_{t_{j+1}})\right\}$. Then, the obtained initial shadow set $\left\{T\left(x_{1}\right),T(x_{2}),\;\ldots ,{T(x}_{t_{j}})\right\}$ is combined to gain $\left\{T\left(x_{1}\right),T(x_{2}),\;\ldots ,{T(x}_{t_{j+1}})\right\}.\;$Next, $\left\{T\left(x_{1}\right),T(x_{2}),\;\ldots ,{T(x}_{t_{j+1}})\right\}$ is used to restore the secret images$\;{SI}_{j+1}^{\prime }$. The XOR operation is performed on the (*j* + 1)th essential shadow set to obtain the (*j* + 1)th cover shadows.The value of *j* is added to 1, and if *j* is less than *i*, step 2 is repeated.The restored rearranged secret images ${SI}_{1}^{\prime },{SI}_{2}^{\prime },\ldots ,{SI}_{i}^{\prime }$ use the inverse of the invertible rearranging matrix during encoding to obtain the distortion-free secret images ${SI}_{1},\;{SI}_{2},\ldots ,{SI}_{i}$.

In (*t*_1_, *t*_2_, …, *t_r_*, *k*, *n*)-essential multiple-secret image sharing, when the union of all essential shadows is obtained, as well as the $k-t_{r}\;$non-essential shadows, the restoration algorithm of the secret image *SI*_1_, *SI*_2_, …, *SI_r_*_+1_ is as follows:

Input: The union of all essential shadow sets $\left\{E_{1,1},\;E_{1,2},\ldots ,\;E_{1,t_{1}}\right\}\cup \left\{E_{2,t_{1}+1},\;E_{2,t_{1}+2},\ldots ,\;E_{2{,t}_{2}}\right\}\cup \ldots \cup \left\{E_{r,t_{r-1}+1},\;E_{r,t_{r-1}+2},\ldots ,\;E_{r,t_{r}}\right\}$ with each shadow size $\left(\frac{rw}{t_{1}+t_{2}+\ldots +t_{r}+k}\right)\times w$ and any $k-t_{r}\;shadows\;of\;$the non-essential shadow set $\left\{{NE}_{1},\;NE_{2},\ldots ,\;{NE}_{n-t_{r}}\right\}$.

Output: The secret images *SI*_1_, *SI*_2_, …, *SI_r_*_+1_, each of size *w × w*.

The steps are as follows:The first shadow set $\left\{E_{1,1},\;E_{1,2},\ldots ,\;E_{1,t_{1}}\right\}$, i.e., the initial shadow set $\left\{T\left(x_{1}\right),T(x_{2}),\;\ldots ,{T(x}_{t_{1}})\right\}$, is used to restore the rearranged secret image, ${SI}_{1}^{\prime }$. The XOR operation is performed on all shadows in the initial shadow set to obtain the first cover shadow. Let *j* = 1.The XOR operation is performed on the *j*th cover shadow with the (*j* + 1)th essential shadow set $\left\{E_{j+1,t_{j}+1},\;E_{j+1,t_{j}+2},\ldots ,\;E_{j+1{,t}_{j+1}}\right\}$ to obtain the initial shadow set $\left\{T(x_{t_{j}+1}),T(x_{t_{j}+2}),\;\ldots ,{T(x}_{t_{j+1}})\right\}$. Then, the obtained initial shadow sets $\left\{T\left(x_{1}\right),T(x_{2}),\;\ldots ,{T(x}_{t_{j}})\right\}$ are combined. Next, $\left\{T\left(x_{1}\right),T(x_{2}),\;\ldots ,{T(x}_{t_{j+1}})\right\}$ is used to restore the secret images$\;{SI}_{j+1}^{\prime }$. The XOR operation is performed on the (*j* + 1)th essential shadow set to obtain the (*j* + 1)th cover shadows.The value of *j* is added to 1, and if *j* is less than *r*, step 2 is repeated.The XOR operation is performed on the *r*th cover shadow with any $k-t_{r}\;shadows\;of$ the non-essential shadow set $\left\{{NE}_{1},\;NE_{2},\ldots ,\;{NE}_{n-t_{r}}\right\},$ $W.L.O.G.\;saying\;\left\{{NE}_{1},\;NE_{2},\ldots ,\;{NE}_{k-t_{r}}\right\}$ to obtain the corresponding subset$\;W.L.O.G.\;saying\;\left\{T(x_{t_{r}+1}),T(x_{t_{r}+2}),\;\ldots ,{T(x}_{k})\right\}\;of\;k-t_{r}\;shadows$ in the initial shadow set $\left\{T(x_{t_{r}+1}),T(x_{t_{r}+2}),\;\ldots ,{T(x}_{n})\right\}$. Then, this is combined with the previously obtained initial shadow set $\left\{T\left(x_{1}\right),T(x_{2}),\;\ldots ,{T(x}_{t_{r}})\right\}\;to\;gain\;\left\{T\left(x_{1}\right),T(x_{2}),\;\ldots ,{T(x}_{k})\right\}$. Next, $\left\{T\left(x_{1}\right),T(x_{2}),\;\ldots ,{T(x}_{k})\right\}$ is used to restore the secret images ${SI}_{r+1}^{\prime }$.The restored rearranged secret images ${SI}_{1}^{\prime },{SI}_{2}^{\prime },\ldots ,{SI}_{r+1}^{\prime }$ use the inverse of the invertible rearranging matrix during encoding to obtain the distortion-free secret images ${SI}_{1},\;{SI}_{2},\ldots ,{SI}_{r+1}$.

In the design of the (*t*_1_, *t*_2_, …, *t_r_*, *k*, *n*)-multi-secret essential secret image sharing mechanism, the essential shadow is completely non-tolerant, and only the non-essential shadow is fault-tolerant. Because essential shadows often represent a higher level of important information, non-fault tolerance means that the secrets can only be reconstructed if the conditions are fully met, such as the sufficient and correct participation of the shadow holder. Such restrictions are effective in preventing information leakage or rebuilding under false authorization, ensuring that high-level information is only available under strict authorization. For an attacker, if the essential shadows are not fault-tolerant, even if some of the essential shadows are obtained, the secret image cannot be restored. This is equivalent to increasing the difficulty and cost of cracking high-end secret content, as well as enhancing the defense of the overall mechanism.

## 4. Experimental Results

Figure 1 illustrates the experimental results of the (2, 4, 6, 8)-essential multi-secret image sharing mechanism with the sensor data. Figure 1a is a secret thermal image (Image1) captured by a thermal sensor. Figure 1b shows the recoverable image obtained by rearranging Image1 with the reversible rearrangement matrix $\left[\begin{array}{cc} 30 & 29\\ 31 & 29 \end{array}\right]$. Figure 1c is a secret fingerprint (Image2) captured by a fingerprint sensor. Figure 1d shows the recoverable image obtained by rearranging Image2 with the same reversible rearrangement matrix $\left[\begin{array}{cc} 30 & 29\\ 31 & 29 \end{array}\right]$. Figure 1e is a secret MRI image (Image3) taken by a medical imaging sensor (an MRI scanner) and marked with abnormal areas (e.g., lesions). Figure 1f shows the recoverable image obtained by rearranging Image3 with the reversible rearrangement matrix $\left[\begin{array}{cc} 30 & 29\\ 31 & 29 \end{array}\right]$. Subsequently, Figure 1g,h shows the level 1 essential shadows, Figure 1i,j shows the level 2 essential shadows, and Figure 1k–n shows the non-essential shadows, all generated by implementing the (2, 4, 6, 8)-essential multi-secret image sharing procedure in Figure 1b,d,f. Figure 1o is the rearranged thermal image obtained by restoring all the essential shadows of the first level, while Figure 1p is the recovered thermal secret image obtained by using the inverse rearrangement matrix $\left[\begin{array}{cc} 30 & -29\\ -31 & 30 \end{array}\right]\;$to rearrange Figure 1o. Figure 1q is the rearranged fingerprint image obtained by restoring all the essential shadows of the first and second levels, while Figure 1r is the recovered fingerprint secret image obtained by using the inverse rearrangement matrix $\left[\begin{array}{cc} 30 & -29\\ -31 & 30 \end{array}\right]\;$to rearrange Figure 1q.

Figure 1s is the rearranged MRI image obtained by restoring all the essential shadows of the first and second levels and any two of the non-essential shadows, while Figure 1t is the recovered MRI secret image obtained by using the inverse rearrangement matrix $\left[\begin{array}{cc} 30 & -29\\ -31 & 30 \end{array}\right]$ to rearrange Figure 1s. Notably, the proposed recovery process restores the original MRI classified image with the anomalous area markers intact.

In the above figure, as the proposed method ensures exact pixel-level recovery through reversible transformations and deterministic reconstruction, the restored secret images are bitwise identical to the original secret images. This lossless nature leads to a mean squared error (MSE) of zero, resulting in a PSNR that tends toward infinity and a structural similarity index (SSIM) of 1.0. These outcomes confirm that the reconstructed images are distortion-free.

As essential multi-secret image sharing is a novel approach, there are no articles on the topic. In comparison with essential secret sharing schemes [25], the proposed method benefits from staged recovery with multiple security levels. However, in comparison with a multi-secret sharing scheme, for example, that in [26], the proposed essential multi-secret sharing scheme provides the partial and fault-tolerant recovery of secret images rather than completely no fault tolerance and no progressive recovery, as in [26].

In the essential multi-secret image sharing scheme proposed in this paper, a method is designed to share multiple secret sensor images, and multiple secret sensor images can be restored according to the obtained essential or non-essential shadows. Multiple secret sensor images can be restored in stages, which can create specific application scenarios for image security.

## 5. Security Analysis

In this section, the proposed (*t*_1_, *t*_2_, …, *tᵣ*, *k*, *n*)-essential multi-secret image sharing scheme is analyzed from a security perspective, focusing on common threat models such as shadow interception, collusion, and unauthorized reconstruction.

(1)Shadow interception

Essential shadows are strictly non-fault-tolerant, which means that, unless the complete set of required essential shadows for a given level is obtained, the corresponding secret image cannot be reconstructed. Intercepting fewer essential shadows yields no partial recovery due to the threshold enforcement and a lack of redundancy.

(2)Collusion attacks

Even if multiple parties collude and combine their non-essential shadows or partial sets of essential shadows, the structure of the polynomial-based encoding prevents successful interpolation unless the number of required linearly independent shares is satisfied. This property ensures that incomplete collaboration does not lead to data leakage.

(3)Rearrangement obfuscation

All secret images are obfuscated using a reversible rearrangement matrix before encoding. This transformation eliminates spatial coherence in the original images, rendering the shadows visually meaningless and resistant to visual pattern attacks or heuristic analysis.

(4)Unauthorized reconstruction

The scheme’s hierarchical design ensures that lower-level secrets may be restored only with their corresponding level of essential shadows. Higher-level secrets are cryptographically protected and inaccessible without the complete higher-level shadow sets. This enforces access control and limits information exposure in the case of partial compromise.

Through these mechanisms, the proposed scheme exhibits strong resilience against passive eavesdropping, shadow interception, and active collusion, making it suitable for secure, context-aware image distribution applications in real-world environments.

## 6. Performance Consideration and Comparative Analysis

(1)Computational complexity:

The proposed EMSIS scheme involves two main computational stages: (1) pixel rearrangement using an invertible matrix, and (2) polynomial-based encoding for shadow generation. The rearrangement process operates in linear time with respect to the number of pixels *O*(*w*^2^), and the polynomial encoding per sector of *t* pixels and *n* shadow holders has complexity *O*(*nt*). All XOR operations are low-cost and can be efficiently computed. As the image is segmented into sectors, these operations are highly parallelizable, making the scheme scalable for large sensor images.

(2)Scalability:

The design is modular and allows for parallel processing per sector and per image level. This means that high-resolution images (e.g., 1024 × 1024) can be split and processed in parallel threads or distributed environments, such as edge servers or GPUs, thereby reducing the total encoding and decoding time.

(3)Storage and transmission overhead:

Each generated shadow represents only a portion of the image data and is not a full duplication of the image. For *r*+1 secret images with *s* essential and *n−s* non-essential shadows, the total storage requirement is approximately proportional to (*r*+1)*w*^2^, which increases linearly with the number of images and shadow count. This redundancy is deliberately introduced to provide fault tolerance and access control. In deployment, only the minimum required shadows need to be stored or transmitted, balancing efficiency and security.

To clarify the differences and advantages of the proposed EMSIS scheme over traditional essential and multi-secret image sharing methods, a comparative summary is provided in Table 1.

The results demonstrate that the proposed method provides unique advantages in flexibility, security, and staged image reconstruction, which are not supported by earlier approaches. While all schemes support lossless recovery, only EMSIS supports multi-level access, selective restoration, and context integration.

## 7. Conclusions

In this study, we proposed a (*t*_1_, *t*_2_, …, *t_r_*, *k*, *n*)-essential multi-secret image sharing mechanism integrated with sensor data and reversible rearrangement matrices to ensure the secure sharing and accurate reconstruction of sensitive images. The experimental results validate the effectiveness of the proposed approach. Secret images from various sensors—including thermal, fingerprint, and medical MRI scanners—were successfully rearranged using a reversible transformation, partitioned into essential and non-essential shadows, and subsequently recovered according to their designated security levels. Importantly, the restoration process preserved critical information, such as the anomalous region markers in MRI images, without distortion or information loss. These results demonstrate that the proposed method provides a robust, flexible, and fault-tolerant solution for multi-level essential image sharing, maintaining the high fidelity of the original secret images while accommodating the varying importance and security requirements of different types of sensor data.

## Figures and Tables

**Figure 1 jimaging-11-00228-f001:**
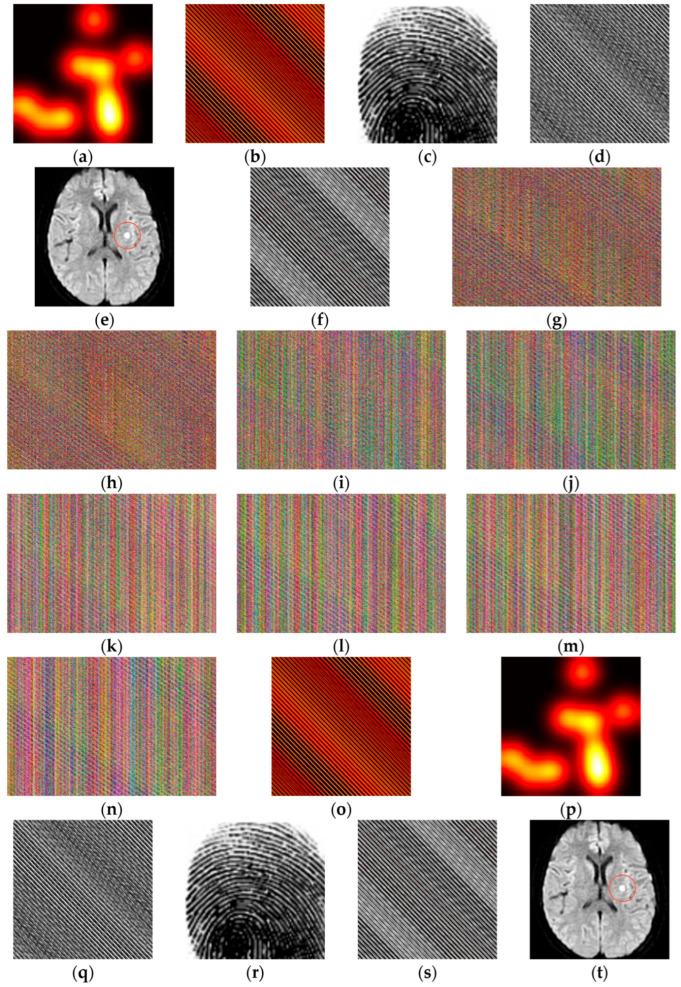
(2, 4, 6, 8)—Experimental results of essential multi-secret image sharing: (**a**) secret thermal image Image1, (**b**) rearranged image of Image1, (**c**) secret fingerprint image Image2, (**d**) rearranged image of Image2, (**e**) secret MRI image of Image3, (**f**) rearranged image of Image3, (**g**,**h**) first-leveled essential shadows, (**i**,**j**) second-leveled essential shadows, (**k**–**n**) non-essential shadows, (**o**) recovered by rearranged images (**g**,**h**), (**p**) recovered images by rearranging (**o**), (**q**) recovered by rearranged images (**g**–**j**), (**r**) recovered images by rearranging (**q**), (**s**) recovered by rearranged images (**g**–**j**) and any two rearranged images (**k**–**n**), and (**t**) recovered images by panel (**s**).

**Table 1 jimaging-11-00228-t001:** Comparative summary of the proposed EMSIS scheme, an essential secret image sharing method [25], and a multi-secret image sharing method [26].

Feature/Method	Proposed EMSIS Scheme	(*t*, *s*, *k*, *n*)-ESS [25]	DNA-Multi-SIS [26]
Recovery type	Lossless	Lossless	Lossless
Hierarchical recovery	Yes	No	No
Fault tolerance (non-essential shadows)	Yes	Limited	No
Shadow size uniformity	Adjustable	Fixed	Fixed
Context-aware reconstruction (e.g., sensor data)	Yes	No	No
Resistance to partial shadow attack	High	Moderate	Low

## Data Availability

The reader who wants to get the images in the paper can send email to cotachen@saturn.yzu.edu.tw.

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
