# Peer review of "Essential Multi-Secret Image Sharing for Sensor Images"

_2313-433X, 2025, doi:10.3390/jimaging11070228_

Round 1
Reviewer 1 Report
Comments and Suggestions for Authors
The authors propose a multi-secret image sharing mechanism. Additionally, the authors explain that the system utilizes sensor data to implement context-based access control, categorizes secret images into hierarchical levels to ensure fault-tolerance, uses a reversible image reflow matrix to protect the images, and allows image restoration to be adjusted through a gradual and flexible restoration mechanism.
Here are some comments:
- While the paper claims that secret images are restored without distortion, there is no mention of any quantitative evaluation to support this. Authors need to provide quantitative evaluations, such as PSNR or SSIM, to validate the accuracy of the restored images.
- While the proposed method emphasizes security, there is limited discussion on potential attack scenarios or threat models. Authors need to elaborate on how the scheme protects against shadow interception or unauthorized reconstruction.
- The algorithm and restoration accuracy are well explained, but how were performance issues (computational complexity, scalability, etc.) considered when handling large sensor images? In addition, what about the storage and transmission overhead due to shadow generation and sharing?
- The paper mentions advantages over existing methods, but there is no experimental comparison. Authors need to provide quantitative or tabular results showing differences in recovery performance, fault tolerance, etc.
Author Response
Comment1: While the paper claims that secret images are restored without distortion, there is no mention of any quantitative evaluation to support this. Authors need to provide quantitative evaluations, such as PSNR or SSIM, to validate the accuracy of the restored images.
Response1: Thanks for reviewer’s insightful comment. In the proposed method, the reconstruction process is strictly lossless, meaning that each pixel in the restored image is exactly identical to the original.
Due to this exact pixel-level recovery, the Mean Squared Error (MSE) between the original and the restored image is zero, leading to a theoretically infinite PSNR value. Similarly, the SSIM score is exactly 1.0, reflecting perfect structural similarity.
To clarify this, the author has added a statement in Section 4 of the revised manuscript explaining that the proposed method guarantees lossless reconstruction by design, and that this can be interpreted as PSNR → ∞ and SSIM = 1.0.
Comment2: While the proposed method emphasizes security, there is limited discussion on potential attack scenarios or threat models. Authors need to elaborate on how the scheme protects against shadow interception or unauthorized reconstruction.
Response2: Thanks for reviewer’s constructive feedback. The author agrees that a thorough discussion on potential attack scenarios and threat models is essential to highlight the security robustness of the proposed scheme.
In response, A new section have been added in the revised manuscript (Section 5: Security Analysis) that outlines common threat models, including shadow interception, shadow collusion, and unauthorized reconstruction. The design addresses these concerns:
The essential shadows are non-fault-tolerant and hierarchically structured, meaning that even if some are intercepted, they cannot be used to reconstruct high-level secret images unless all required essential shadows are correctly obtained.
The use of reversible rearrangement matrices obscures the spatial structure of the images, making partial shadows unintelligible and resisting visual cryptographic attacks.
Polynomial-based encoding ensures that without a sufficient number of linearly independent shadows, the attacker cannot reconstruct the secret due to the hardness of polynomial interpolation.
The phased recovery mechanism and shadow-level separation limit the information gain from partial shadow sets, preventing information leakage from lower-level reconstructions.
These additions clarify the scheme’s resilience against both passive eavesdropping and active collusion attacks.
Comment3: The algorithm and restoration accuracy are well explained, but how were performance issues (computational complexity, scalability, etc.) considered when handling large sensor images? In addition, what about the storage and transmission overhead due to shadow generation and sharing?
Response3: Thanks for the reviewer’s insightful and important question regarding the performance and overhead of the proposed scheme.
In response, a Performance Consideration subsection has been added in the revised manuscript to address computational complexity, scalability, and storage/transmission overhead.
Computational Complexity and Scalability:
The proposed scheme operates on pixel-wise transformations using reversible rearrangement and polynomial-based encoding per sector. The rearrangement matrix is a linear-time operation O(w2), and the encoding per sector is based on polynomial evaluation and XOR, with time complexity O(nt), where n is the number of shadow holders and t is the number of pixels per sector. These operations are parallelizable and scale linearly with the image size. For large-scale sensor images (e.g., MRI), the system can process image blocks in parallel using GPU or multicore hardware.
Storage and Transmission Overhead:
The author acknowledges that generating multiple essential and non-essential shadows introduces redundancy. However, the design ensures that:
- Each shadow is of size comparable to the original image sector (not the full image)
- Non-essential shadows provide fault tolerance and partial reconstruction ability, which is critical in sensor networks prone to data loss.
In practical deployment, only the required number of essential shadows (and a few optional non-essential ones) need to be transmitted or stored per access policy, optimizing the trade-off between redundancy and robustness.
These considerations are now clearly described in the new Section 6 of the manuscript.
Comment4: The paper mentions advantages over existing methods, but there is no experimental comparison. Authors need to provide quantitative or tabular results showing differences in recovery performance, fault tolerance, etc.
Response4: Thanks for reviewer’s valuable comment. The author agrees that a comparative evaluation with existing methods can strengthen the credibility of the proposed scheme.
In the revised manuscript, a comparative analysis table is added to highlight the differences in key aspects such as recovery performance, fault tolerance, and hierarchical reconstruction capability between the proposed method and representative schemes from the literature, specifically:
- (t, s, k, n)-Essential Secret Image Sharing (e.g., [25])
- General Multi-Secret Image Sharing (e.g., [26])
Due to the fundamental differences in design goals and shadow structure, direct pixel-level recovery accuracy comparison is not always meaningful. However, the author focuses on structural features such as:
- Lossless recovery guarantee
- Support for hierarchical/staged reconstruction
- Fault tolerance in shadow transmission
- Context-aware reconstruction
The author has included this comparison in the new Section 6 as a new table (Table 1) to offer a clearer view of the relative advantages of the proposed EMSIS scheme.
Reviewer 2 Report
Comments and Suggestions for Authors
The author proposes a new essential multi-secret image sharing scheme. However, the scope of the study appears to be quite limited. The literature review is insufficient and mostly references outdated work. There has been significant progress in this field, especially since 2020, which is not reflected in the current manuscript. Additionally, the paper lacks a proper background or preparatory section to contextualize the proposed scheme. Potential attacks should be considered. Security and performance analysis should be included. So, I think the paper does not meet the standard required for publication.
Author Response
Comment: The author proposes a new essential multi-secret image sharing scheme. However, the scope of the study appears to be quite limited. The literature review is insufficient and mostly references outdated work. There has been significant progress in this field, especially since 2020, which is not reflected in the current manuscript. Additionally, the paper lacks a proper background or preparatory section to contextualize the proposed scheme. Potential attacks should be considered. Security and performance analysis should be included. So, I think the paper does not meet the standard required for publication.
Response: The author sincerely appreciates detailed and critical comments from the reviewer. The author also fully acknowledges that the earlier version of the manuscript lacked a comprehensive literature review, contextual background, and detailed analysis of security and performance. Based on the suggestions of the reviewer, the manuscript has significantly revised as follows:
- Expanded and Updated Literature Review:
The author has revised the introduction and added a new section in Section 2 to cover recent advances in the field, especially between 2022 and 2025. Notable additions include:
- Yadav et al. (2022) on meaningful essential secret image sharing with uniform shares.
- Wang et al. (2022) on multi-level progressive sharing with hierarchical shadows.
- Dehkordi et al. (2024) on verifiable essential secret sharing using LWE-based cryptography.
- Chattopadhyay et al. (2025) on DNA-encrypted verifiable multi-secret sharing.
These recent works are now clearly cited and contrasted with the proposed EMSIS scheme to better highlight our method’s novelty in hierarchical access control and sensor-aware integration.
- Added a “Related Works and Motivation” Section:
A new section in Section 2 is introduced that explains the conceptual evolution from traditional secret image sharing to essential sharing, and the need for multi-level, context-aware schemes in sensor-driven environments (e.g., healthcare, battlefield, IoT). This section positions EMSIS as a response to the limitations of earlier fixed-threshold or non-progressive methods.
- Security Analysis Section Included:
As per the reviewer’s suggestion, a full Security Analysis section is added in Section 5. It includes:
- Threat modeling of shadow interception and collusion attacks
- Protection mechanisms via non-fault-tolerant essential shadows
- The role of rearrangement matrix in obscuring visual content
- Resistance to partial reconstruction through incomplete sets
This helps to substantiate the robustness of our scheme under practical adversarial assumptions.
- Performance Evaluation Added:
A dedicated section titled Performance Consideration and Comparative Analysis in Section 6 is included, analyzing:
- Computational complexity (O(nt) per sector with linear scaling)
- Scalability via parallel processing
- Storage and transmission overhead from essential and non-essential shadows
- Trade-offs between redundancy and fault tolerance
- Comparative Table with Prior Methods:
To concretely demonstrate the relative strengths of EMSIS, a comparative table in Section 6 contrasting the proposed scheme with existing ones is added in terms of:
- Lossless recovery
- Fault tolerance
- Shadow size control
- Context-aware features
- Hierarchical reconstruction
This helps clarify the advantages of the proposed method beyond conceptual description.
Hope these substantial revisions address the reviewer’s concerns and bring the manuscript up to a level suitable for publication. The author greatly values the reviewer’s feedback and guidance in improving the quality and completeness of the work.
Round 2
Reviewer 1 Report
Comments and Suggestions for Authors
All the requested revisions have been properly addressed, and the quality of the manuscript has improved. I believe the paper is now suitable for publication.
Reviewer 2 Report
Comments and Suggestions for Authors
The authors have revised the manuscript as requested:
- Expanded and Updated Literature Review
- Added a “Related Works and Motivation” Section
- Security Analysis Section Included
- Performance Evaluation Added
- Comparative Table with Prior Methods
The paper can be accepted in its current form.